# Clinical Impact of the *CYP2C19* Gene on Diazepam for the Management of Alcohol Withdrawal Syndrome

**DOI:** 10.3390/jpm13020285

**Published:** 2023-02-03

**Authors:** Teresa T. Ho, Melissa Noble, Bao Anh Tran, Katlynd Sunjic, Sheeba Varghese Gupta, Jacques Turgeon, Rustin D. Crutchley

**Affiliations:** 1Department of Pharmacotherapeutics & Clinical Research, University of South Florida Taneja College of Pharmacy, Tampa, FL 33612, USA; 2Department of Pharmaceutical Sciences, University of South Florida College of Pharmacy, Tampa, FL 33612, USA; 3Precision Pharmacotherapy Research & Development Institute, Tabula Rasa HealthCare, Moorestown, NJ 08057, USA; 4Department of Pharmacotherapy, Washington State University, College of Pharmacy and Pharmaceutical Sciences, Yakima, WA 98901, USA

**Keywords:** diazepam, benzodiazepines, pharmacogenomics, alcohol withdrawal syndrome, personalized medicine, CYP2C19

## Abstract

Diazepam is a benzodiazepine widely prescribed for the management of patients with severe alcohol withdrawal syndrome to prevent agitation, withdrawal seizures, and delirium tremens. Despite standard dosing of diazepam, a subset of patients experience refractory withdrawal syndromes or adverse drug reactions, such as impaired motor coordination, dizziness, and slurred speech. The CYP2C19 and CYP3A4 enzymes play a key role in the biotransformation of diazepam. Given the highly polymorphic nature of the *CYP2C19* gene, we reviewed the clinical impact of variants in the *CYP2C19* gene on both the pharmacokinetics of diazepam and treatment outcomes related to the management of alcohol withdrawal syndrome.

## 1. Introduction

The 2019 National Survey on Drug Use and Health reported that 14.5 million people were classified as having an alcohol use disorder (AUD), with an estimated 65 million Americans engaging in some type of recent alcohol binge [1]. Alcohol use disorders can include a combination of physical and behavioral symptoms, including withdrawal, tolerance, and craving [2]. The cessation of alcohol after heavy (more than eight drinks per week for women and more than 15 drinks per week for men) and prolonged use can lead to alcohol withdrawal syndrome (AWS) [2]. Symptoms of alcohol withdrawal can develop from hours to days after cessation [2,3].

Chronic alcohol intake causes downregulation of the γ-aminobutyric acid (GABA) inhibitory receptors and upregulation of the N-methyl-D-aspartate (NMDA) excitatory receptors. Therefore, when there is an abrupt cessation of alcohol consumption, neuronal hyperactivity develops due to overactivation of the NMDA receptors and impaired inhibitory activity from the downregulation of the GABA receptors [4]. Signs and symptoms of alcohol withdrawal may include diaphoresis, tremors, nausea, vomiting, tachycardia, hypertension, hyperthermia, anxiety, sleep disturbances, and hallucinations. Appropriate and timely management of alcohol withdrawal is imperative to prevent the most serious complications, such as seizures, delirium, and death, from occurring [5,6].

According to the American Society of Addiction Medicine (ASAM) Clinical Practice Guideline on Alcohol Withdrawal Management, benzodiazepines are recommended as a first-line treatment due to the agents’ cross-tolerance with alcohol and more favorable side effect profile than the use of alcohol as a treatment itself [5,7]. Benzodiazepines bind to and activate the gamma-aminobutyric acid type A (GABA_A_) receptors, leading to an influx of intracellular chloride, which causes central nervous system depression and subsequently reduces symptoms and incidences of seizures and delirium [5]. Alcohol itself, if used for alcohol withdrawal syndrome, may still impair the function of the liver, pancreas, and bone marrow [7]. Other agents have demonstrated their place in the treatment of AWS (alpha-2-agonists, beta-antagonists, carbamazepine, and neuroleptics) but are not recommended as monotherapy [7]. Benzodiazepines do have potential for drug–drug interactions with other CNS depressants, including alcohol, if use is continued, and patients and providers should remain aware of this risk [5]. The risk of benzodiazepine dependence increases over time; thus, dosage and duration should be limited to what is required for the indication [8]. The most commonly used benzodiazepines for the management of AWS include diazepam, lorazepam, and chlordiazepoxide [5]. For a smoother withdrawal, long-acting benzodiazepines, such as diazepam and chlordiazepoxide, are recommended [5,9]. Different dosing methods may be utilized for the treatment of AWS based on severity; these may include fixed dosing, front loading, and symptom-triggered dosing [5]. The severity of alcohol withdrawal is evaluated based on a series of presenting symptoms using the Clinical Institute Withdrawal Assessment for Alcohol revised (CIWA-Ar). CIWA-AR scores of less than 10, 10 to 18, and 19 or greater define mild, moderate, and severe AWS, respectively [5].

Patients with resistant AWS, defined as severe or complicated alcohol withdrawal signs and symptoms, despite administering high benzodiazepine doses, have a higher incidence of intubation, longer intensive care unit (ICU) stays, and an increased risk of nosocomial infections compared to patients responsive to benzodiazepines [10,11]. Given that adverse drug reactions (ADRs) may occur with exceedingly high doses of diazepam in patients with resistant AWS and inter-individual variation in diazepam dose requirements exists, a personalized medicine approach may help optimize the use of diazepam [12,13]. Diazepam is predominantly metabolized by CYP2C19 and CYP3A4, both of which are subject to genetic variations that alter metabolic activity [8]. Herein, this review article provides an overview of diazepam and its role in the management of AWS, summarizes the impact of *CYP2C19* variants on diazepam pharmacokinetics and AWS outcomes during treatment with diazepam, and concludes with suggestions for additional future research to determine the clinical utility of genotype-guided testing for diazepam.

## 2. Diazepam Pharmacokinetics and Pharmacogenetics

Diazepam is commercially available in oral, intravenous, intramuscular, intranasal, and rectal formulations. Following oral administration of diazepam, more than 90% is absorbed, and peak plasma concentrations are achieved in 1–1.5 h [8]. Diazepam is highly lipophilic, highly protein bound, and has a volume of distribution of 0.8 to 1.0 L/kg [8]. Diazepam is metabolized by CYP450 enzymes into three active metabolites: nordiazepam, oxazepam, and temazepam (Figure 1). The major active metabolite, nordiazepam, is produced by the N-demethylation of diazepam. N-demethylation is catalyzed by both CYP3A4 and CYP2C19. Nordiazepam undergoes 3-hydroxylation via CYP3A4 to form oxazepam. Temazepam, another active metabolite of diazepam, also undergoes subsequent metabolic transformation to R and S isomers of oxazepam in the presence of CYP3A4 and CYP2C19 [14]. (R, S)-oxazepam undergoes phase II metabolism via glucuronidation in the presence of uridine diphosphate glucuronosyltransferase (UGT) isoforms. S-oxazepam is glucuronidated by UGT2B15, whereas UGT1A9 and UGT2B7 are responsible for the glucuronidation of R-oxazepam. The rate of glucuronidation of S-oxazepam is affected by *UGT2B15* variants and the gender of the subject [15].

The *CYP2C19* gene is highly polymorphic, with more than 35 genetic variants known to influence the activity of the CYP2C19 enzyme [17]. The frequencies of these *CYP2C19* alleles vary across ancestrally diverse populations and may be found on the PharmGKB Gene-specific information tables for *CYP2C19* [18]. The most common loss-of-function alleles include the *CYP2C19**2 (c.681G>A; rs4244285) and *CYP2C19**3 alleles (c.636G>A; rs4986893), which lead to a premature stop codon and a non-functional truncated protein [18]. Individuals that are carriers of one or two loss-of-function alleles are predicted to have intermediate or poor CYP2C19 metabolism, respectively [19]. The allele frequency of the *CYP2C19**2 allele is around 15% in Caucasians and Africans, and 29–35% in Asians [20]. Other *CYP2C19* variant alleles, including *CYP2C19**3, are present in less than 1% of most populations. However, among Asians, the allele frequency of *CYP2C19**3 ranges between 2 and 9%. Increased CYP2C19 enzyme activity results from a variant in the promoter region that enhances the transcription of the gene and is defined by the *CYP2C19**17 allele (c.-806C>T; rs12248560) [20]. On average, the *CYP2C19**17 allele frequency ranges between 3 and 21% [21]. Individuals who are carriers of one or two increased function alleles are predicted to have rapid or ultrarapid CYP2C19 activity, respectively [18]. If no genetic variants are detected in laboratory analyses, the wild-type *CYP2C19**1 allele is assigned. Although this allele is associated with normal enzyme activity, it is important to note that it is still possible that an individual may have rare variants in *CYP2C19* that were not detected during analysis. 

While the activity of CYP3A4 is highly variable, genetic variants account for a minor proportion of this variability. Several variants have been identified in *CYP3A4* [22]. A frequently reported variant with functional significance in the literature is the *CYP3A4**22 allele (c.522-191 C>T; rs35599367), which results in decreased CYP3A4 activity [23]. The minor allele frequency of the *CYP3A4**22 allele is 5% in Europeans, 3% in admixed Americans, and is rare in other populations [24].

## 3. Diazepam for Management of Alcohol Withdrawal Syndrome

Benzodiazepines are generally considered the preferred first-line treatment strategy due to their well-established benefit in reducing signs and symptoms of withdrawal, including seizures and delirium. Benzodiazepines, like alcohol, activate the GABA receptors, which leads to inhibitory effects. Therefore, benzodiazepines are considered cross tolerant with alcohol as they replace the inhibitory effects of alcohol [6,7]. Although other agents, such as gabapentin, carbamazepine, topiramate, and baclofen, may have utility in managing alcohol withdrawal, benzodiazepines are the only agents, with well-documented efficacy in preventing delirium tremens, seizures, and mortality [6]. Diazepam, chlordiazepoxide and lorazepam are the benzodiazepines that have been most studied and utilized in practice for managing alcohol withdrawal [5,7]. No current data support the superior efficacy of one agent over another. Because of this, benzodiazepine selection is typically based on pharmacokinetic properties, including onset and duration of action [7]. Benzodiazepines with longer durations of action are commonly preferred by providers because the prolonged half-life leads to a smoother course of withdrawal and minimizes the risk of rebound symptoms and seizures [5,7]. The duration of the clinical effects is determined by the half-life of the parent drug and active metabolites. Diazepam has the longest half-life, 20–80 h and 30–100 h for the active metabolites, and the shortest time to peak effect of the benzodiazepines typically considered for alcohol withdrawal management, making it an ideal treatment option [6,25]. The use of shorter-acting agents, such as lorazepam, may be associated with increased rates of symptom re-emergence, anxiety, tachycardia, and seizures [6].

A variety of dosing strategies, including symptom-triggered dosing, fixed dosing, and front loading, have been described as potential treatment approaches for the management of AWS [5]. A symptom-triggered regimen involves only administering medication when symptoms cross a pre-specified severity threshold based on a structured assessment scale. The CIWA-Ar is such an assessment, composed of 10 items, mostly scored as 0–7, with a maximum score of 67. The medication dosage can also be adjusted based on the severity of the symptoms, as assessed by the CIWA-Ar (0–8, withdrawal absent to minimal; 9–15, moderate withdrawal; ≥16, severe withdrawal) [5]. In a fixed dosing strategy, a pre-determined dose of the medication is administered on a set schedule with the dose decreased gradually over a few days. Finally, with front loading, a moderate to high dose of a long-acting agent is administered frequently at the beginning of treatment to quickly control signs and symptoms of withdrawal. Drug levels are then allowed to self-taper through metabolism [5].

Symptom-triggered dosing is the preferred treatment strategy for alcohol withdrawal as it is associated with a shorter duration of treatment, less medication administered, and reduced inpatient length of stay compared to a fixed dosing strategy [5,7]. However, in patients at high risk for developing severe alcohol withdrawal, a front-loading approach should be utilized due to evidence suggesting a reduction in treatment duration, occurrence of seizures, and delirium duration. Diazepam is the preferred benzodiazepine for this dosing strategy to achieve rapid control of the symptoms and then allow for a gradual taper [5]. In patients with moderate to severe alcohol withdrawal, intravenous (IV) administration of benzodiazepines, such as diazepam and lorazepam, may be preferred to rapidly control symptoms and prevent progression [6]. Diazepam and lorazepam IV are also considered to be first-line treatment options for patients experiencing alcohol withdrawal seizures due to their rapid onset of action [5]. Diazepam is more lipophilic than lorazepam, allowing for more diffusion across the blood–brain barrier. As such, IV diazepam’s onset of action is 5 min compared to 30 min with IV lorazepam. Oral diazepam also has the fastest onset of action of the oral benzodiazepines used for alcohol withdrawal [6].

However, it is important to note that longer-acting benzodiazepines, such as diazepam, are more likely to accumulate and lead to possible respiratory depression and oversedation, which may present as confusion, memory issues, ataxia, and delirium. Those with hepatic dysfunction and older patients are most at risk of this accumulation [5]. Some data suggest that these risks can be reduced by adjusting the dose based on the individualized patient response [6]. Additionally, repeated administration of IV diazepam can be associated with a risk of hyponatremia and metabolic acidosis because the medication is solubilized in propylene glycol [5]. Due to the rapid onset of action with IV diazepam, it is also imperative to monitor patients at a heightened risk for respiratory depression [5].

## 4. Impact of *CYP2C19* Variants on Diazepam Pharmacokinetics

Previous studies have demonstrated that genetic variation in the *CYP2C19* gene impacts diazepam pharmacokinetics and the potential patient response to diazepam. In a pharmacokinetic study of 21 healthy Chinese males, the plasma elimination half-lives of diazepam and desmethyldiazepam were nearly three- and two-times longer among CYP2C19 poor metabolizers (100.8 ± 32.3 h and 219.9 ± 62.7 h, respectively) compared to CYP2C19 normal metabolizers (34.7 ± 23 h and 103.1 ± 25.9 h, respectively) following administration of diazepam (5 mg orally) [26]. These pharmacokinetic differences have also been corroborated in other populations, including Japanese and Swedish subjects [27,28,29]. Among CYP2C19 rapid or ultrarapid metabolizers, a study evaluating steady-state concentrations of diazepam in AWS patients found that *CYP2C19**17 variant allele carriers (89.12 (53.26; 178.07) ng/mL) had significantly lower levels of diazepam compared to CYP2C19 normal metabolizers (250.70 (213.34; 308.53) ng/mL) [30,31].

## 5. Impact of *CYP2C19* Variants on Adverse Effects, Efficacy, and Dosing of Diazepam

The pharmacokinetic differences in diazepam based on the *CYP2C19* genotype may impact adverse drug events and treatment efficacy of diazepam. In a prospective study investigating the effects of *CYP2C19* variants on recovery from general anesthesia with diazepam 0.1 mg/kg IV and sevoflurane maintenance among 63 Japanese patients scheduled for a mastectomy or leg surgery, emergence times were significantly longer for CYP2C19 intermediate (median 13 min) and poor metabolizers (median 18 min) compared to CYP2C19 normal metabolizers (median 10 min) [26]. While the clinical significance of the observed recovery times remains unclear based on the *CYP2C19* genotype, extreme duration of sedation was reported in a 77-year-old male treated with diazepam 10 mg IV receiving a cumulative dose of 400 mg of diazepam to achieve continuous sedation within 54 h for treatment of alcohol withdrawal and delirium [32]. The patient required 18 days of flumazenil administration to reverse the effects of diazepam before regaining consciousness. Retrospective genotyping revealed that the patient was a CYP2C19 intermediate metabolizer (*CYP2C19**1/*2). In addition to the *CYP2C19* genotype, other factors thought to contribute to prolonged sedation in this patient’s case included advanced age, hypoalbuminemia, and hepatic impairment. Despite reports of CYP2C19 poor metabolizers having higher plasma diazepam concentrations than CYP2C19 normal metabolizers, a study of 15 healthy Japanese participants, following a single dose of diazepam 5 mg IV, did not find significant differences in sedative effects, as assessed by psychomotor tests among the different *CYP2C19* phenotype groups [26,33,34,35,36].

One research group in Moscow evaluated differences in diazepam efficacy in the management of AWS based on *CYP2C19* genotype. In a prospective cohort study, 30 Russian males with a CIWA-AR score greater than 10 were administered a fixed dose of diazepam 10 mg IM every eight hours for five days [37]. Changes in CIWA-AR scores from day 1 to day 5 of treatment were greater among patients who were carriers of the *CYP2C19**2 variant allele (−12 (−13, −9)) compared to those who were CYP2C19 normal metabolizers (−8.5 (−15, −5)), indicating diazepam may be more effective in carriers of the *CYP2C19**2 allele compared to CYP2C19 normal metabolizers [37]. CYP2C19 normal (−12 (−15, −8)) metabolizers had a greater change in CIWA-AR scores from day 1 to day 5 of treatment with diazepam compared to those who were carriers of the *CYP2C19**17 variant allele (−7 (−14, −5)). These findings potentially indicate that in the management of AWS patients, diazepam treatment may be less effective in individuals with increased CYP2C19 activity compared to those with normal CYP2C19 activity. To evaluate the side effects associated with diazepam by the *CYP2C19* genotype, investigators used the Udvalg for Kliniske Undersøgelser (UKU) side effect rating scale, as assessed by a healthcare professional, which comprises three parts, including 48 items (grouped under psychic, neurological, autonomic, and other), a global assessment of side effects on daily performance, and a statement addressing action taken to address the side effects. Differences in the UKU score before and after treatment with diazepam were significantly greater among carriers of the *CYP2C19**2 allele (9.5 (8.0; 11.0)) demonstrating a higher risk of adverse events compared to CYP2C19 normal metabolizers (7.0 (6.0; 12.0), *p* = 0.009) [37]. In comparison to CYP2C19 normal metabolizers (8.0 (6.0; 12.0)), the difference in the UKU score before and after treatment was significantly less among carriers of the *CYP2C19**17 allele (6.0 [6.0; 12.0], *p* = 0.006) indicating a lower risk of adverse events [37]. When the investigators expanded the sample size from 30 to 50 and 100 Russian males in subsequent investigations and correlated outcomes with therapeutic drug monitoring, similar differences in CIWA-AR and UKU score changes were found [30,31].

In contrast, a prospective cohort study of 69 South Indian males found no significant differences in the diazepam loading dose requirements, time required for reversal of acute symptoms, or frequency of persistent symptoms following successful treatment of an acute episode of AWS based on the *CYP2C19* genotype [38]. Study participants were given a loading dose of diazepam 10 mg IV initially, followed by diazepam 20 mg orally if the CIWA-AR score remained at more than eight at the end of each of the two hourly assessments. A successful loading dose of diazepam was defined as the patient attaining a drowsy but arousable state, or if the patient had two consecutive CIWA-AR values assessed two hours apart that were eight or less. The absence of an association between diazepam and *CYP2C19* in this cohort compared to recently published studies may be due to several factors, including being underpowered and the median baseline CIWA-AR score of 14 (moderate) being notably lower than the baseline CIWA-AR score of 21 (severe) in the aforementioned studies [38]. Table 1 summarizes studies evaluating the impact of the *CYP2C19* genotype on the efficacy and safety of diazepam in patients with AWS.

## 6. Role of Pharmacogenomic Testing for Diazepam

Based on studies evaluating the association between *CYP2C19* and diazepam among AWS patients, reduced CYP2C19 activity may lead to increased concentrations of diazepam, therefore placing these patients at a higher risk of experiencing adverse effects. Conversely, increased CYP2C19 activity may result in reduced concentrations of diazepam, leading to treatment failure with diazepam and theoretically higher dose requirements or alternative treatments for the management of AWS. The FDA-approved package labeling for diazepam acknowledges that compounds inhibiting CYP3A4 and CYP2C19 may lead to increased and prolonged sedation, making it plausible that carriers of the loss-of-function allele in *CYP2C19* may also be adversely affected [39]. In light of recent publications, the impact of the *CYP2C19* gain-of-function allele on the efficacy of diazepam may also need to be considered in the management of AWS but is not currently addressed in the FDA labeling of diazepam. The Clinical Pharmacogenomics Implementation Consortium (CPIC) assigns a CPIC level for prescribing actionability from A, indicating that a drug may be guided by genetic information, to D, indicating weak evidence for genetic information guiding a drug [40]. For diazepam, a provisional CPIC level of B/C denotes that the evidence requires further review before prescribing actionability can be determined. The Pharmacogenomics Knowledgebase (PharmGKB) clinically annotates variant drug pairs with levels of evidence ranging from level 1 (high evidence) to level 4 (unsupported), with diazepam rated as level 3 for its association with the *CYP2C19**2 and *CYP2C19**3 variants [41].

The literature to date describing the clinical utility of genes involved in the pharmacokinetics and pharmacodynamics of chlordiazepoxide and lorazepam is limited. A preliminary suggestion for future investigation of a pre-emptive pharmacogenomic guided approach to AWS might include considering use of alternative first line benzodiazepine agents chlordiazepoxide or lorazepam in clinical scenarios where the *CYP2C19* genotype predicts poor efficacy or an increased risk of adverse effects with diazepam. Another area that deserves further investigation is whether increased dosing of diazepam is necessary to help manage AWS in individuals who are CYP2C19 rapid and ultrarapid metabolizers. Pre-emptive *CYP2C19* genotype-guided diazepam has been implemented in the perioperative setting but has not yet been studied in the management of AWS [42]. The clinical significance of the observed pharmacokinetic changes in diazepam based on *CYP2C19* variants requires further investigation in prospective *CYP2C19* genotype-guided diazepam studies for AWS treatment and should take into account route of administration, dosing approach (front loading versus fixed schedule versus symptom triggered), and gender. Consideration may also be given to evaluating the combinatorial impact of both the *CYP2C19* and *CYP3A4* genes on diazepam, given two studies by Skryabin et al. demonstrated a potential association between genetic variation in *CYP3A4* and the safety and efficacy of diazepam in managing patients with AWS, for which additional studies are also needed to confirm findings [43,44].

## 7. Conclusions

Rapid identification and management of AWS is critical for relieving and preventing the symptoms and complications of AWS. Diazepam is a first-line benzodiazepine agent used to treat AWS. Recent studies evaluating the clinical impact of the *CYP2C19* gene on diazepam for AWS have identified that individuals with reduced CYP2C19 activity may be at increased risk of adverse effects whereas those with increased CYP2C19 activity may experience reduced efficacy with standard doses of diazepam. However, additional research is warranted to determine the clinical utility of *CYP2C19* genotype-guided testing for diazepam to improve overall treatment outcomes in individuals who experience AWS.

## Figures and Tables

**Figure 1 jpm-13-00285-f001:**
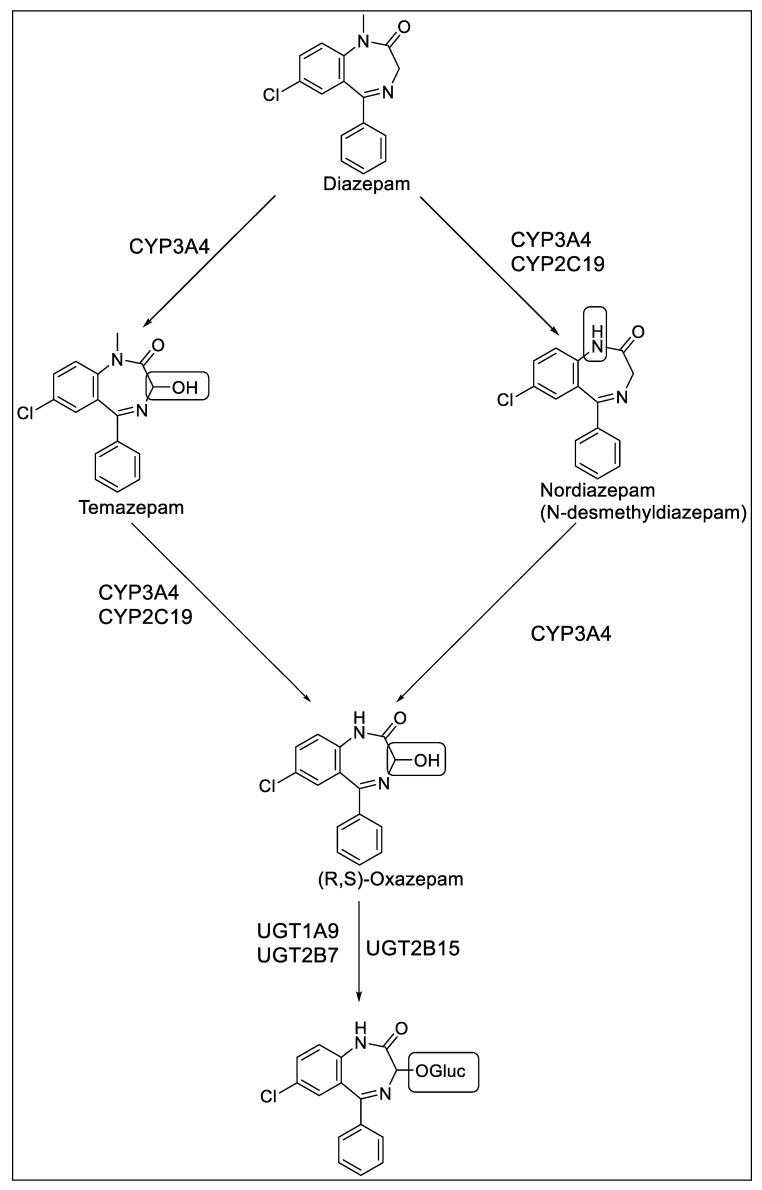
Diazepam metabolic pathway [16].

**Table 1 jpm-13-00285-t001:** Summary of studies evaluating the impact of *CYP2C19* genotype on the efficacy and safety of diazepam.

Author, Year	Study Design	Objective	Study Population	CYP2C19 Genotype or Phenotype Distribution	Diazepam Dosing Strategy	Efficacy and/or Safety Results
Garmen et al., 2015 [32]	Case report	Report on outcome of AWS patient treated with diazepam	77-year-old male with AWS and delirium	CYP2C19 IM (*CYP2C19*1/*2)*	10 mg IV (received cumulative dose of 400 mg to achieve continuous sedation within 54 h)	Extreme sedation requiring 18 days of flumazenil
Jose et al., 2016 [38]	Prospective cohort study	Evaluate the effect of *CYP2C19* variants on diazepam loadingdose requirement and time to reversal of AWS	69 South Indian patients with CIWA-AR ≥ 10	CYP2C19 UM (*1/*17, *17/*17): 18 CYP2C19 NM (*1/*1): 9 CYP2C19 IM (*1/*2, *1/*3, *2/*17): 32 CYP2C19 PM (*2/*2, *2/*3: 10	10 mg IV loading dose followed by 20 mg PO if CIWA-AR ≥ 8 at end of each two hourly assessments	No association identified between *CYP2C19* variants and loading dose requirements or time to reversal of AWS
Skryabin et al., 2020 [37]	Prospective cohort study	Evaluate the effect of *CYP2C19**2 and *CYP2C19**17 on the efficacy and safety of diazepam in patients with AWS	30 Russian patients with CIWA-AR > 10	*CYP2C19**1/*1: 22 *CYP2C19**1/*2 or *CYP2C19**2/*2: 8	10 mg IM every eight hours for 5 days	Change in CIWA-AR score from day 1 to day 5: *CYP2C19**1/*1, −8.5 [−15.0; −5.0] vs CYP2C19*2 allele carriers, −12.0 [−13.0; −9.0], *p* = 0.021 Change in UKU score from day 1 to day 5: *CYP2C19**1/*1, 7.0 [6.0; 12.0] vs. *CYP2C19**2 allele carriers, 9.5 [8.0; 11.0], *p* = 0.009 Change in CIWA-AR score from day 1 to day 5: *CYP2C19**1/*1, −12.0 [−15.0; −8.0] vs. *CYP2C19**17 allele carriers, −7.0 [−14.0; −5.0], *p* < 0.001 Change in UKU score from day 1 to day 5: *CYP2C19**1/*1, 8.0 [6.0; 12.0] vs. *CYP2C19**17 allele carriers, 6.0 [6.0; 12.0], *p* < 0.001
				*CYP2C19**1/*1: 17 *CYP2C19**1/*17 or *CYP2C19**17/*17: 13		
Skryabin et al., 2021 [30]	Prospective cohort study	Evaluate the effects of *CYP2C19**17 on the steady-state concentration of diazepam in patients with AWS as well as impact on efficacy and safety	50 Russian patients with CIWA-AR > 10	*CYP2C19**1/*1: 30 *CYP2C19**1/*17: 16 *CYP2C19**17/*17: 4	10 mg IM every eight hours for 5 days	Change in CIWA-AR score from day 1 to day 5: *CYP2C19**1/*1, −12.0 [−15.0; −8.0] vs. *CYP2C19**17 allele carriers, −7.0 [−14.0; −5.0], *p* < 0.001 Change in UKU score from day 1 to day 5: *CYP2C19**1/*1, 8.0 [6.0; 12.0] vs. *CYP2C19**17 allele carriers, 6.0 [6.0; 12.0], *p* < 0.001
Skryabin et al., 2022 [31]	Prospective cohort study	Evaluate the effects of *CYP2C19**17 on plasma and saliva concentrations of diazepam as well as impact on efficacy and safety	100 Russian patients with CIWA-AR > 10	*CYP2C19**1/*1: 65 *CYP2C19**1/*17 or *CYP2C19**17/*17: 35	10 mg IM every eight hours for 5 days	Patients with the *CYP2C19**1/*17 and *CYP2C19**17/*17 genotype had a smaller difference in the CIWA-AR (reduced diazepam efficacy) scores and slower increase in the UKU score (lower risk of adverse effects) compared to patients with the *CYP2C19**1/*1 genotype from day 1 to day 6

AWS, alcohol withdrawal syndrome; CIWA-AR, Clinical Institute Withdrawal Assessment for Alcohol, revised; UM, ultrarapid metabolizer; NM, normal metabolizer; IM, intermediate metabolizer; PM, poor metabolizer; UVU, udvalg for kliniske undersøgelser.

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
