# Peer review of "Clinical Impact of the CYP2C19 Gene on Diazepam for the Management of Alcohol Withdrawal Syndrome"

_jpm, 2023, doi:10.3390/jpm13020285_

Round 1

Reviewer 1 Report

Clinical impact of CYP2C19 gene on diazepam for management of alcohol withdrawal syndrome

Overall comments:

This is a well written manuscript. However, I would not describe it as a review, more likely a commentary. For a review, even if descriptive, the authors need to provide a methods section pointing out how the literature was selected and reviewed (search terms, exclusion/inclusion criteria of papers, etc). In addition, the section on genetics is short and could do with broadening; From a basic search (2000 – 2023) of the terms “alcohol withdrawal”, ‘benzodiazepines’ on Pubmed , multiple (1000+) manuscripts were obtained.  A table of papers discussed either sorted by 2C19 * allele or by phenotype (PM, IM NM, RM, UM) and resultant PK effect should be included. As per my comments below at minimum CYP3A4 and CYP3A5 should also be included.

Specific comments:

·       Authors mention that alcohol binging can down-regulate GABA receptors which is where BZDs act. How then are they effective if the receptors they work on are down-regulated or internalized? – perhaps a follow-up after lines 47/48 would be helpful.

·       BZDs themselves can cause dependance/addiction – the authors should include a few lines discussing the pros/cons of using BZDs for AWS.

·       The CYP3A4 gene also has genetic variants which impact on BZDs. In addition CYP3A5 can impact on Diazepam – see:

o   https://pubmed.ncbi.nlm.nih.gov/35471917/

o   https://pubmed.ncbi.nlm.nih.gov/33622083/

o   The authors should clarify why only 2C19 was focused on (lines 67-69) – in my opinion 3A4/5 should be included.

o   Other genes which may impact on BZD disposition are NAT2 and UGT2B15, see:

§  https://www.pharmgkb.org/pathway/PA165111375

o   Lines 102 -105 are not factually correct based on the literature available.

·       Lines 164 -173 – Units are missing from the number values provided.

·       Lines 174 – 194 seem irrelevant as this is not in patients with AWS – I would suggest deletion and only including literature relevant to AWS.

·       Line 197- CIWA-AR needs to be defined and explained.

Reviewer 2 Report

This is a review exploring the role of pharmacokinetic variation in patients with resistant alcohol withdrawal syndrome, defined as severe or complicated alcohol withdrawal signs and symptoms despite administration of high benzodiazepine doses. Specifically, they focused on CYP2C19 and CYP3A4, which are the main isoenzymes metabolizing diazepam, the treatment of choice for the management of AWS.

The review is generally well written, but I have some suggestions:

- Search methodology should be described, and prioritization of evidence should be provided

- The section on clinical recommendations is too vague, a box/table would help

- If loss of function or inhibition of CYP3A and or CYP2C19 leads to improved sedation it should be also described what happens when the isoenzymes increase their function (i.e. ultra metabolizers) 

Reviewer 3 Report

The review performed by Ho et al. is a summary of the main findings regarding CYP2C19 pharmacogenetics on diazepam use for alcohol withdrawal syndrome (AWS). Mainly, I found the manuscript fluid and easy to understand. I have some comments I would like to address for the authors:

1. Why did the authors choose to review only CYP2C19 and not CYP3A4 as well?

2. The authors have justified the use of diazepam in AWS, however, I would like to know if there are any pharmacogenetic studies in AWS regarding other benzodiazepines, considering the adverse effects they are known to cause.

3. In the paragraph the authors cite the CYP2C19 variants (lines 84-105) they present some references regarding the variant frequency in some major population groups. I suggest the authors use a public genomic database (i.e. gnomAD) to retrieve and present the variants' frequency across the populations. I believe a table with this data would benefit the manuscript.

4. I suggest the authors alter the terms "polymorphisms" to "variants", as the latest HGVS recommendation.

5. Minor point: in line 222, there is a misspelling in CYP3A4.

Round 2

Reviewer 1 Report

The authors have improved the manuscript by making changes suggested by multiple reviewers.

My main issue still stands at the methodology. A review is a comprehensive piece of work. The article still does not outline how the review was conducted. Stating that "a pubmed search was carried out is not sufficient".  I urge the authors to read prior reviews to see what is required. See: https://pubmed.ncbi.nlm.nih.gov/26328136/
